# FULL-LENGTH MRNA DESIGN WITH REWARD-GUIDED MASKED DIFFUSION MODEL FINE-TUNING

**Sawan Patel**[*], **& Sherwood Yao** [†]
Atom Bioworks
Cary, NC 27513, USA
{s.patel,s.yao}@atombioworks.com

**Sophia Tang** [*]
Department of Computer and Information Science
University of Pennsylvania
Philadelphia, PA 19104, USA
{sophtang}@seas.upenn.edu

**Pranam Chatterjee** [†]
Department of Computer and Information Science
Department of Bioengineering
University of Pennsylvania
Philadelphia, PA 19104, USA
{pranam}@seas.upenn.edu

## ABSTRACT

High-fitness mRNA design requires the simultaneous optimization of multiple application-critical properties, but the design principles governing this high-dimensional landscape remain poorly understood. This motivates the need for methods that provide systematic, controllable generation of mRNA sequences. To address this, we introduce **T3PO-mRNA**, a framework for computing reward-guided discrete diffusion trajectories that iteratively construct increasingly accurate approximations of the Pareto frontier. Our approach leverages tree search to identify high-reward sequence trajectories and uses these trajectories to fine-tune diffusion models on progressively stronger sequence buffers. We demonstrate that **T3PO-mRNA** effectively designs therapeutic mRNAs with optimized half-life and translation efficacy, enabling both improved multi-objective performance and efficient inference-time sampling over prior inference-time guidance methods.

## 1 INTRODUCTION

Messenger RNA (mRNA) pharmacueticals have shown to be effective treatment options for combating disease (Polack et al., 2020; Schuster et al., 2019; Baden et al., 2021; Xu et al., 2024). Much effort is being directed towards extending mRNA therapeutics to other afflictions, but controllably durable protein expression is required for any application involving mRNA to be clinically viable (Parhiz et al., 2024). The consideration of therapeutically-relevant properties during the mRNA design process, such as translation efficiency and intracellular half-life, is difficult in practice, as manipulating any component of the mRNA can have pronounced effects on overall fitness (Li et al., 2025). Though careful perturbation of individual components within an mRNA sequence, such as at the 5' (Li et al., 2025) or 3'UTR (Li et al., 2025; Ma et al., 2025), has been explored, full-sequence mRNA optimization is comparably unexplored. Additionally, expression profiles can vary widely depending on the tissue or cell line of interest (Zheng et al., 2025).

Deep learning methods have previously been explored for mRNA property prediction and generation (Zhang et al., 2024; Chu et al., 2024; Barazandeh et al., 2025; Castillo-Hair et al., 2024), though these are typically limited to individual components within the mRNA. Notably, masked diffusion models and inference-time multi-property guidance (Patel et al., 2025) have shown capability for conditional full-length mRNA design. However, the likelihood of producing suboptimal mRNAs with a pretrained masked diffusion model is likely when exposed primarily to poor-fitness regions with high data density.

---

[*]Equal contribution.
[†]Corresponding author.

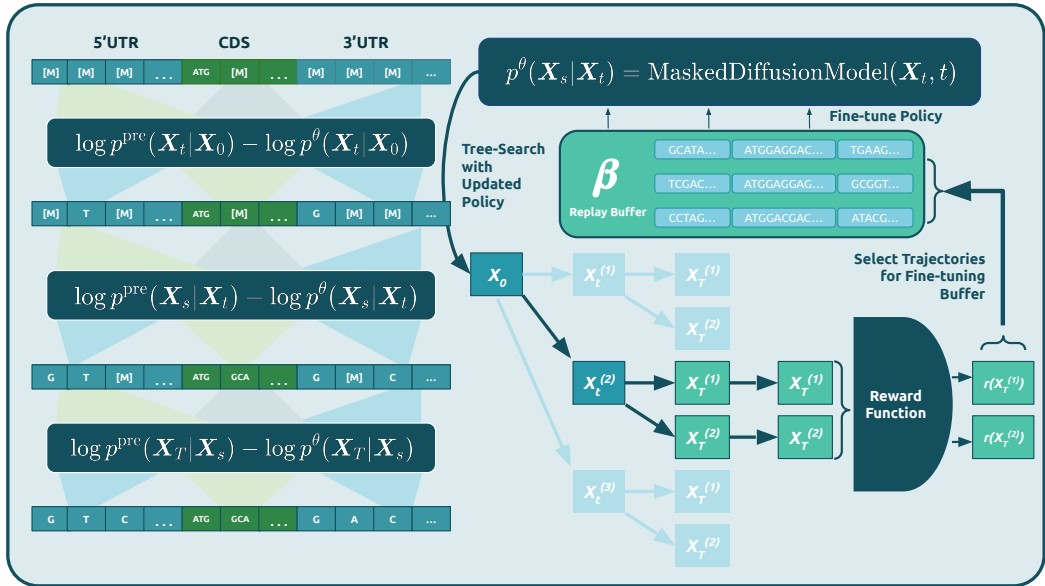

Figure 1: **T3PO-mRNA**. A method for reward-guided, trajectory-aware fine-tuning of an mRNA masked diffusion model for pareto optimal mRNA design.

Here, we introduce **T**ree Search Guided **T**rajectory-Aware Fine-**T**uning for **P**arallel **O**ptimality - **mRNA** (**T3PO-mRNA**), a framework for computing reward-guided discrete diffusion trajectories that iteratively construct increasingly accurate approximations of the Pareto frontier. We demonstrate that **T3PO-mRNA** is capable of producing mRNAs with heightened intracellular half-life and translation efficiency relative to sensible baselines *in silico* and over existing inference-time guidance methods.

## 2 METHODS

### 2.1 TREE SEARCH GUIDED TRAJECTORY-AWARE FINE-TUNING

We implement the TR2-D2 algorithm for reward-guided mRNA design (Tang et al., 2025b) using mRNAutilus (Patel et al., 2025). This method integrates off-policy reinforcement learning (RL) paired with a Monte Carlo Tree Search algorithm to procure high-reward trajectories for fine-tuning a pretrained generative model. The full algorithm is given in Algorithm 1.

**Entropy-Regularized Reward Optimization of Masked Diffusion Models** The pre-trained masked diffusion model generates a path measure of discrete diffusion trajectories $\mathbb{P}^{\text{pre}}$ over a time horizon $t \in [0, T]$ that samples from the data distribution $\mathbb{P}^{\text{pre}} = p_{\text{data}}$. Given a reward function $r(\boldsymbol{X}_T) : \mathbb{R}^d \to \mathbb{R}$, reward-guided fine-tuning aims to sample from the **reward-tilted distribution**:

$$\mathbb{P}^{\star}(\boldsymbol{X}_{0:T}) := \frac{1}{Z} \mathbb{P}^{\text{pre}}(\boldsymbol{X}_{0:T}) \exp\left(\frac{r(\boldsymbol{X}_T)}{\alpha}\right), \quad \mathbb{P}_T^{\star}(\boldsymbol{X}_T) \propto p_{\text{data}}(\boldsymbol{X}_T) \exp\left(\frac{r(\boldsymbol{X}_T)}{\alpha}\right) \quad (1)$$

which is the solution to the **entropy-regularized reward optimization** problem (Uehara et al., 2024):

$$\max_{\theta} \mathbb{E}_{\boldsymbol{X}_{0:T} \sim \mathbb{P}^{\theta}}[r(\boldsymbol{X}_T)] - \alpha D_{\text{KL}}(\mathbb{P}^{\theta} \| \mathbb{P}^{\text{pre}}) \quad (2)$$

where $\mathbb{P}^{\theta}$ is the path measure generated by the fine-tuned model parameters $\theta$ and $\mathbb{P}^{\text{pre}}$ is the frozen pre-trained model. The KL divergence term ensures that the fine-tuned model doesn't diverge too far from the pre-trained path measure, which is controlled by the hyperparameter $\alpha$.

**Discrete Diffusion Fine-Tuning with Off-Policy Reinforcement Learning** To solve the entropy-regularized reward optimization problem in (2) and obtain an optimal fine-tuned discrete diffusion

model $\mathbb{P}^\theta = \mathbb{P}^\star$, we leverage the off-policy RL algorithm introduced in TR2-D2 (Tang et al., 2025b). Specifically, we use the **weighted denoising cross-entropy** (WDCE) loss given by:

$$\mathcal{L}_{\text{WDCE}} := \mathbb{E}_{\boldsymbol{X}_{0:T} \sim \mathbb{P}^\star}[\mathcal{L}_{\text{DCE}(\theta;\boldsymbol{x})}] = \mathbb{E}_{\boldsymbol{X}_{0:T} \sim \mathbb{P}^{\bar\theta}}\left[\underbrace{\frac{d\mathbb{P}^\star}{d\mathbb{P}^{\bar\theta}}}_{\exp(w^\star(\boldsymbol{X}_T))} \mathcal{L}_{\text{DCE}}(\theta;\boldsymbol{x})\right] \tag{3}$$

where $\mathbb{P}^{\bar\theta}$ denotes the non-gradient tracking fine-tuned model from a previous iteration $\bar\theta = \texttt{stopgrad}(\theta)$ used to generate the replay buffer and $w^\star(\boldsymbol{X}_T)$ is the importance weight assigned to the clean sequence $\boldsymbol{X}_T$ that measures how optimal it is under the entropy-regularized reward-maximization problem (2):

$$\log \frac{d\mathbb{P}^\star}{d\mathbb{P}^\theta}(\boldsymbol{X}_{0:T}) = \underbrace{\frac{r(\boldsymbol{X}_T)}{\alpha} + \sum_{t:\boldsymbol{X}_s \neq \boldsymbol{X}_t} \sum_{\ell:\boldsymbol{X}_s^\ell \neq \boldsymbol{X}_t^\ell} \log \frac{p^{\text{pre}}(\boldsymbol{X}_s^\ell | \boldsymbol{X}_t^{\text{UM}})}{p^\theta(\boldsymbol{X}_s^\ell | \boldsymbol{X}_t^{\text{UM}})} - \log Z}_{w^\star(\boldsymbol{X})} \tag{4}$$

which assigns higher weight to sequences $\boldsymbol{X}_T = \boldsymbol{x}$ that have high reward $r(\boldsymbol{x})$ and low divergence in log-probability of the generation trajectory $p_\theta$ from its log-probability under the pre-trained model $p_{\text{pre}}$. The weighted loss $\mathcal{L}_{\text{DCE}}$ is the standard **denoising cross-entropy** (DCE) loss given a clean sample $\boldsymbol{x} \sim p_{\text{target}}$ defined as:

$$\mathcal{L}_{\text{DCE}}(\theta;\boldsymbol{x}) := \mathbb{E}_{t \sim \mathcal{U}(0,1)}\left[\frac{1}{t}\mathbb{E}_{p_t(\tilde{\boldsymbol{x}}_t|\boldsymbol{x})} \sum_{\ell:\tilde{\boldsymbol{x}}_t^\ell = \boldsymbol{M}} -\log p_\theta(\tilde{\boldsymbol{x}}_t^\ell = \boldsymbol{x}^\ell | \tilde{\boldsymbol{x}}_t^{\text{UM}})\right] \tag{5}$$

where $p_\theta(\tilde{\boldsymbol{x}}_t^\ell = \boldsymbol{x}^\ell | \tilde{\boldsymbol{x}}_t^{\text{UM}})$ is the probability of the ground truth token at position $\ell$, denoted $\boldsymbol{x}^\ell$, given the unmasked tokens $\tilde{\boldsymbol{x}}_t^{\text{UM}}$ under the current model $\theta$. In practice, we optimize this objective $\mathcal{L}_{\text{WDCE}}$ by (1) sampling a replay buffer $\mathcal{B} := \{(\boldsymbol{x}, w^\star)\}$ of sequences from the fine-tuned policy while tracking its log-probability under pre-trained model and final reward for computing $w^\star(\boldsymbol{x})$, (2) remasking and computing the DCE loss $\mathcal{L}_{\text{DCE}}$ under the current model $\theta$, and (3) weighting the loss terms and updating the fine-tuned parameters $\theta$ with $\nabla_\theta \mathcal{L}_{\text{WDCE}}$.

**Multi-Objective Optimization** Balancing several reward functions $\boldsymbol{r} = (r_1, \ldots, r_K) : \mathcal{X} \to \mathbb{R}^K$ requires maximizing the tradeoffs across each objective. Instead of iterating towards a single optimal reward value, we define a Pareto frontier denoted $\mathcal{P}^*$ of *reward vectors* where no reward can be further optimized without sacrificing the other rewards. Practically, it suffices to approximate $\mathcal{P}^*$ by exploring the solution space $\mathcal{X}$ and populating a set $\mathcal{P}^*$ if a sequence is non-dominated by any existing sequences in $\mathcal{P}^*$. To do this, we leverage the Monte Carlo Tree Guidance (MCTG) algorithm from Tang et al. (2025a), which iteratively performs **selection** of optimal unmasking steps, **expansion** of diverse unmasking steps, **rollout** to a clean sequence and reward evaluation, and **backpropagation** of rewards to encourage further exploration of high reward steps (Section A.5. During the rollout phase, the masked diffusion model is prompted with a partially-masked token sequence and is constrained to sample tokens with single-base resolution for the 5' and 3'UTRs and tokens with codon-level resolution for the CDS.

When updating the replay buffer with a newly sampled trajectory $(\boldsymbol{x}, w^\star(\boldsymbol{X}))$, we consider the Pareto optimality of its reward vector $\nabla(\boldsymbol{x}) \in \mathbb{R}^K$. At each iteration of MCTG, we compare the $M$ rolled out sequences $\{\boldsymbol{x}^i\}_{i=1}^M$ with the current buffer $\mathcal{B}$ and add it to the buffer if it is **non-dominated** by the sequences already in the buffer.

$$\mathcal{B} \leftarrow \mathcal{B} \cup \{\boldsymbol{x}^i \mid \nexists \tilde{\boldsymbol{x}} \in \mathcal{B} \quad \text{s.t.} \quad \forall k, r_k(\tilde{\boldsymbol{x}}) \geq r_k(\tilde{\boldsymbol{x}}^i) \wedge \exists k, r_k(\tilde{\boldsymbol{x}}) > r_k(\boldsymbol{x}^i)\}$$

While computing the exact Pareto frontier is computationally intractable in practice, each iteration within the search provides a better approximation of the true $\mathcal{P}^*$.

## 2.2 PREDICTING HALF-LIFE AND TRANSLATION EFFICIENCY

We train XGBoost regression models to predict experimentally-collected, therapeutically-relevant properties from sequence embeddings. In this work, we focus on optimizing mRNA sequences for **intracellular half-life** and **translation efficiency in human cells** from the input sequence.

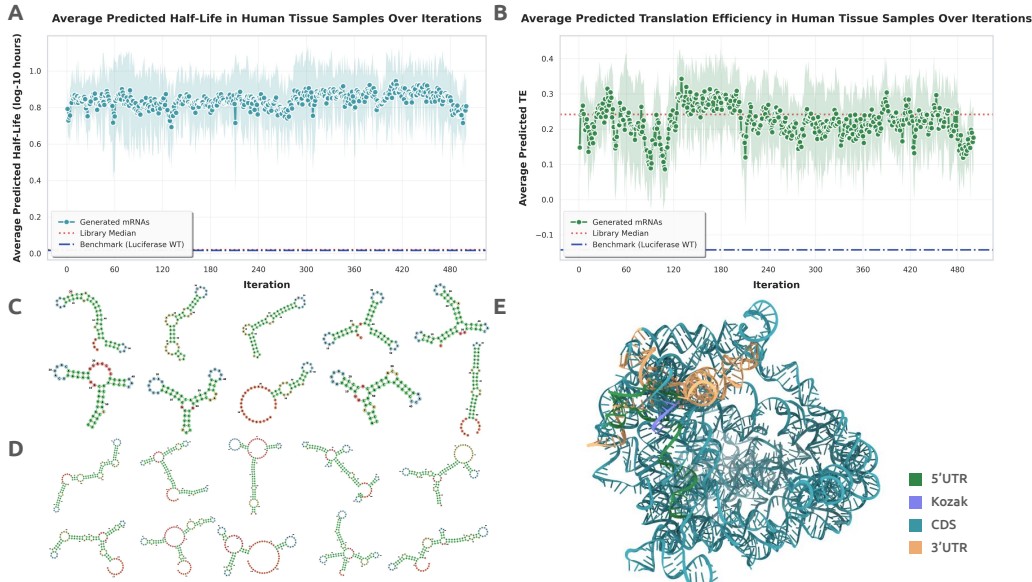

Figure 2: **Conditional generation of full-length *P. pyralis* luficerase mRNAs with reward-guided masked diffusion model fine-tuning.** At each buffer generation step, we generate a pool of 250 sequences optimized for translation efficiency and half-life in human cells for downstream off-policy fine-tuning. Their predicted (**A**) intracellular half-lives and (**B**) translation efficiencies are shown over each fine-tuning epoch, with accompanying standard deviation. Secondary structures for the generated (**C**) 5'UTRs and (**D**) 3'UTRs were predicted and visualized. (**E**) A predicted 3D structure for one of the full-length mRNAs is provided, colored by sequence locality.

**Translation Efficiency**    We use an XGBoost boosted tree (Chen & Guestrin, 2016) regression model trained to predict mRNAutilus embeddings mRNA sequence translation efficiency using mRNAutilus embedded mRNAs using data collected from (Zheng et al., 2025) spanning 1,282 human and 995 mouse datasets across several cell lines (Figure A1A).

**Half-Life**    Similarly to translation efficiency, we use sequence embeddings to train an XGBoost regressor to predict mRNA half-life (log-10). Paired data was collected from (Agarwal & Kelley, 2022), which sourced transcriptome-wide half-life data from human samples across 16 publications and deposits. Overall, it contains data for nearly 13,000 transcripts across 39 human samples. Results are shown in Figure A1B. Additional details for regressor training are in Section A.3.

## 3    RESULTS

### 3.1    *P. Pyralis* LUCIFERASE MRNA DESIGN

**Experimental Setup**    We first designed a library of *P. Pyralis* luciferase mRNAs with T3PO-mRNA. For this task, we prompted mRNAutilus with a template luciferase mRNA sequence consisting of fully-masked 5' and 3'UTRs (lengths 50 and 114, respectively) and an ORF derived from the GenScript F-Luc CDS. The template's ORF spanned 551 codon tokens that, save for the start and stop codons, were uniformly masked at a rate of $0.15$. This template mRNA was then provided as an input to the T3PO-mRNA algorithm for mRNA library design.

At each buffer generation step, we generate a pool of 250 sequences optimized for translation efficiency and half-life in human cells using Monte Carlo Tree Search for downstream off-policy fine-tuning. Hyperparameters are provided in Table A2. Sequences were designed for co-optimality across predicted translation efficiency and half-life in human cells (Section A.3).

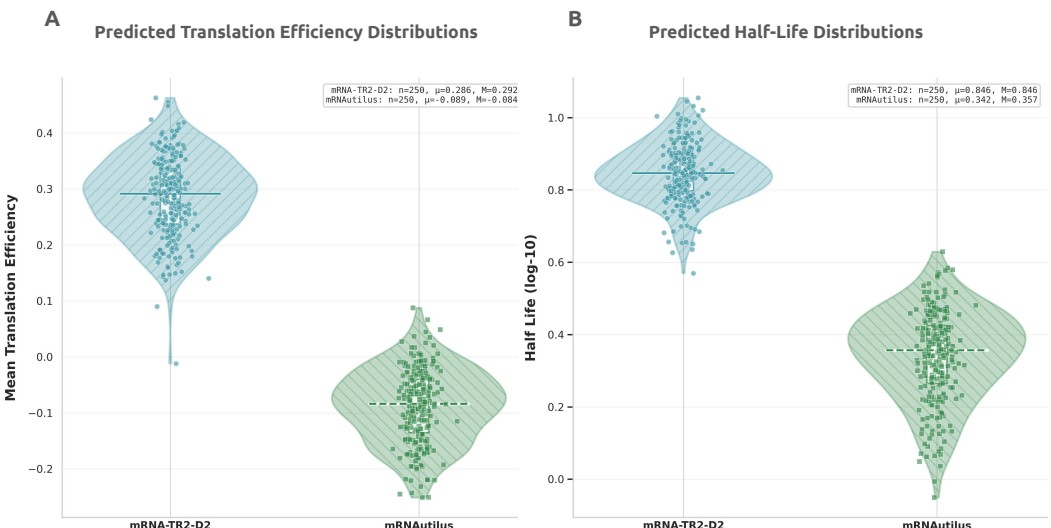

Figure 3: **Evaluation of full-length mRNA design methods.** Libraries of 250 *P. Pyralis* Luciferase mRNAs were produced using T3PO-mRNA and Monte Carlo Tree Guidance, both using the mR-NAutilus diffusion model. Distributions for predicted (**A**) translation efficiency and (**B**) half-life in human cells are shown as violin plots. Swarm plots are embedded within each violin plot.

**Conditional Generation**   We demonstrate that finetuning of the pre-trained mRNAutilus masked diffusion model with **T3PO** yields a pool of high-fitness *P. Pyralis* Luciferase mRNAs *in silico* (Figure 2A, B). For both translation efficiency and intracellular half-life, the mRNA sequences generated from the fine-tuned model score higher than the median mRNAs from the XGBoost regressor training datasets (red line) and a benchmark *P. Pyralis* Luciferase mRNA sequence constructed using the GenScript F-Luc ORF and human alpha-globin UTRs (blue line). We find that the generated 5' and 3'UTRs are highly diverse, as shown from their secondary structures (Figure 2C, D). Furthermore, hierarchical clustering of both sets of UTRs (Figure A2) indicates that designed *P. Pyralis* Luciferase mRNAs are not only highly functional but also structurally diverse. In Figure 2E, we visualize a predicted 3D structure for one of the designed mRNA sequences.

### 3.2   COMPARISON TO INFERENCE-TIME GUIDANCE WITH MONTE CARLO TREE GUIDANCE

To further demonstrate the efficacy of **T3PO-mRNA**, we compare a T3PO-generated library of *P. Pyralis* Luciferase mRNAs to a library produced using inference-time guidance with the MCTG algorithm (Patel et al., 2025; Tang et al., 2025a) with the base masked diffusion model from mRNAutilus (Patel et al., 2025). With both methods, 250 mRNA sequences were optimized for predicted translation efficiency and intracellular half-life in human cells. We show that the library produced by T3PO-mRNA exhibits much higher fitness than the library produced via MCTG, whether for predicted translation efficiency ($+\mathbf{0.375}$; Figure 3A) or half-life ($+\mathbf{0.489}$, 167% increase; Figure 3B).

## 4   DISCUSSION

Therapeutic mRNA design requires consideration of multiple properties which are yet to be fully understood. These properties, including improved stability, reduced toxicity, and prolonged expression, are critical for therapeutic success but often require optimization through careful manipulation of an mRNA sequence and observing its effects *in vitro* or *in vivo*. Designing mRNA sequences with desired properties that can controllably and predictably alter function across any cell line is an ambitious, yet necessary, step for advancing the use of mRNA technologies for therapeutics. Here, we

demonstrate that combining principles from RL, discrete diffusion, and mRNA biology can produce therapeutically-optimized mRNA sequences.

While we have yet to test our method *in vitro*, we find the *in silico* results indicate significant promise. A key limitation is the lack of cell-type-specific property predictors, we emphasize that our method can be adapted for any set of reward predictors in a plug-and-play manner. In contrast to inference-time guidance, fine-tuning introduces the additional cost of modifying the underlying model weights, but this also yields much faster generation since no further iterations or reward evaluations are needed during inference. In total, **T3PO-mRNA** is a practical solution for programmable mRNA design.

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

## A  APPENDIX

### A.1  BASE GENERATIVE MODEL FOR mRNA SEQUENCE DESIGN

**Discrete Diffusion**  Let $\boldsymbol{x} = (\boldsymbol{x}^1, \ldots, \boldsymbol{x}^L) \in \mathcal{V}^L$ denote a token sequence of length $L$ where the vocabulary size is $V$. We denote $p_0(\boldsymbol{x}) = p_{\text{prior}}(\boldsymbol{x})$ as the prior noisy distribution and $p_T(\boldsymbol{x}) = p_{\text{data}}(\boldsymbol{x})$ as the target data distribution, and $p_t(\boldsymbol{x})$ as the time-dependent marginal distribution for $t \in [0, T]$ which interpolates between $p_0$ at $t = 0$ and $p_T$ at $t = T$. Given a noisy sequence $\boldsymbol{x}_0 \sim p_0$ and a clean sequence $\boldsymbol{x}_T \sim p_T$ where $\boldsymbol{x}_T \in \{0, 1\}^V$ is a one-hot vector with 1 in the true state, **discrete diffusion** aims to sample from the clean data distribution $\boldsymbol{x}_T \sim p_{\text{data}}$ given sequences from $p_{\text{prior}}$. To do this, it iteratively samples from a marginal **reverse posterior** $p_{s|t}(\boldsymbol{x}_s|\boldsymbol{x}_t)$ that defines the probability of sampling a slightly less noisy sequence $\boldsymbol{x}_s$ at time $s = t - \frac{1}{T}$, where $T$ is the total number of sampling steps. We define the **conditional reverse posterior** given a clean sequence $\boldsymbol{x}_T$ as $p_{s|t,T}(\boldsymbol{x}_s|\boldsymbol{x}_t, \boldsymbol{x}_T)$. A neural network is trained with parameters $\theta$ to predict the distribution $p_{s|t,T}(\boldsymbol{x}_s|\boldsymbol{x}_t, \boldsymbol{x}_T)$, and sample from the parameterized reverse posterior $p_{s|t}^\theta(\boldsymbol{x}_s|\boldsymbol{x}_t)$ at inference.

**Pre-Trained Masked Discrete Diffusion Model**  We use mRNAutilus (Patel et al., 2025), a pretrained 150M-parameter mRNA masked diffusion model, in the T3PO-mRNA algorithm for iteratively sampling novel sequences within the MCTS algorithm and fine-tuning on the resulting Pareto buffer. Masked diffusion models are simple implementations of discrete diffusion models whereby the prior distribution is chosen as the Dirac distribution concentrated on a fully-masked sequence, or an absorbing state (Sahoo et al., 2024). During the forward process, individual tokens are randomly replaced to mask tokens in a uniform manner based on a chosen noise schedule. The generative process begins from a fully-masked token sequence and iteratively decodes masked tokens back to their original identities according to a parameterized probability distribution conditioned on all unmasked tokens at the preceding step. Additional details on discrete diffusion and mRNA sequence tokenization are given in Sections A.1 and A.2, respectively.

### A.2  TOKENIZATION

We leverage a hybrid tokenization scheme to encode the coding sequence (CDS) and the untranslated regions (UTRs) of an mRNA separately. We tokenize the CDS as 3-mers, effectively by codon. UTRs are separately encoded at single-base resolution. Following this process, the 5'UTR tokens, CDS tokens, and 3'UTR tokens for a given sequence are concatenated to form the full mRNA token sequence. For the CDS, we produce tokens according to the standard codon table, which spans 64 codons, 1 primary start codon, and 3 stop codons. All uracil (U) bases are replaced with thymine (T) to maintain consistency across representations, resulting in a primary UTR vocabulary of four nucleotides: 'A', 'C', 'T', and 'G'. Additionally, we include IUPAC degenerate nucleotide codes such as 'I', 'R', 'Y', 'K', 'M', 'S', 'W', 'B', 'D', 'H', 'V', 'N', and '-' to account for sequence ambiguities introduced by sequencing softwares. For all generated sequences, the Kozak motif ('GCCACC') is appended to the 5'UTR before token unmasking. We include special tokens [CLS], [EOS], [MASK], [UNK], and [PAD], which denote sequence classification, end-of-sequence markers, masked tokens, and padding, respectively. Sequences are truncated or padded to a total of 7,500 tokens for batched training. In total, the vocabulary spans 86 tokens.

### A.3  REGRESSORS

We train XGBoost regressors for mRNA half-life and translation efficiency (Figure A1. Parameters for the XGBoost regressors are listed in Table A1). These regressors are trained to predict a continuous value from token-averaged sequence embeddings.

### A.4  TR2-D2 ALGORITHM

We repurpose the TR2-D2 algorithm for designing Pareto-optimal mRNA sequences as described in Algorithm 1.

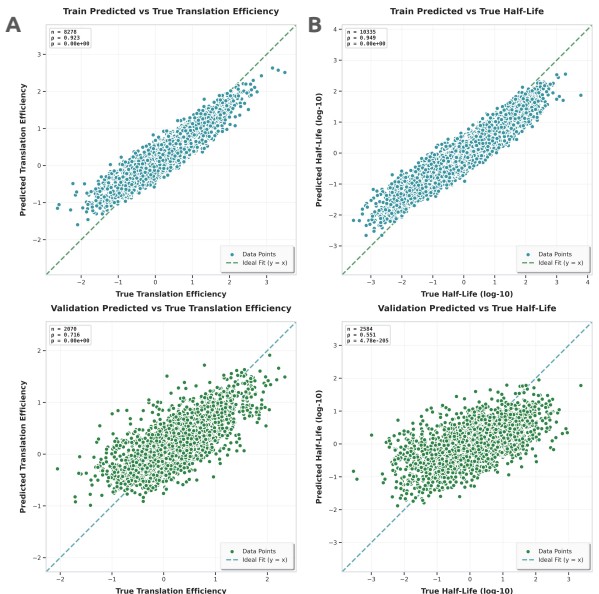

Figure A1: **Gradient-boosted regressors trained on diffusion model embeddings for reward computation.** XGBoost regressors for predicting therapeutically-relevant properties from mRNA sequence embeddings were trained for reward computation. Prediction performance for (**A**) translation efficiency and (**B**) intracellular half-life are shown across the respective training (top) and validation (bottom) sets.

Table A1: XGBoost Hyperparameters for Regression

| Hyperparameter | Value/Range |
|---|---|
| Objective | `reg:squarederror` |
| Lambda | $[0.1, 10.0]$ (log scale) |
| Alpha | $[0.1, 10.0]$ (log scale) |
| Gamma | $[0, 5]$ |
| Colsample by Tree | $[0.5, 1.0]$ |
| Subsample | $[0.6, 0.9]$ |
| Learning Rate | $[1e{-}5, 0.1]$ |
| Max Depth | $[2, 30]$ |
| Min Child Weight | $[1, 20]$ |
| Tree Method | `hist` |
| Scale Pos Weight | $[0.5, 10.0]$ (log scale) |

**Algorithm 1** Enhanced Reinforcement Learning with MCTS-Driven Structured Search for mRNA Design.

1: **Input:** pre-trained model $p^{\text{pre}}$, finetune policy model $p^{u_\theta}$, reward function $r$, prompt $X$, UTR indices $u_1, u_2$, and off-policy RL algorithm
2: Initialize finetune policy model $p^{u_\theta} = p^{\text{pre}}$
3: **while** not converged **do**
4:     **while** buffer $\mathcal{B}$ is not full **do**
5:         Generate samples from current policy model $p^{u_\theta}$ according to prompt $X$ – constrain token selection according to indices $X_i \in \{A, C, T, G\}$ if $i < u_1$ or $i > u_2$ else $X_i \in \{ATG, ATC, \dots TAA\}$
6:         Select optimal samples that maximize the reward function and add to buffer
7:         Explore similar samples given previous selections using the Search algorithm
8:     **end while**
9:     Update $\theta$ using samples from $\mathcal{B}$ using off-policy RL algorithm for multiple epochs
10:     Reset the buffer $\mathcal{B}$
11: **end while**

## A.5 MONTE CARLO TREE GUIDANCE

To guide the unconditional MDM towards Pareto-optimality across multiple properties relevant to mRNA therapeutic efficacy, we leverage the MCTG algorithm for identifying progressively higher-fitness mRNA sequences to populate the fine-tuning buffer (Algorithm 2) (Tang et al., 2025a). Parameters driving the algorithm are shown in Table A2. All other hyperparameters are kept as reported in Tang et al. (2025b).

**Initialization** We initialize a tokenized sequence with a partially masked coding sequence (CDS), masked 5'UTR sequence, and a masked 3'UTR sequence. This becomes the root node of the MCTS tree with timestep $t = 0$. Additionally, we construct a vector scoring function $\mathbf{f} : \mathcal{V}^L \to \mathbb{R}^K$ that returns a score vector given a clean input sequence $x_0 \in \mathcal{V}^L$ and initialize an empty Pareto-optimal set $\mathcal{P}^* = \{\}$. We define the following hyperparameters: number of MCTS iterations $N_{\text{iter}}$ and number of children nodes $N_{\text{child}}$. A template CDS is also provided at input, which is referenced during expansion.

**Selection** To determine the next unmasking step from a parent node with an already expanded set of child sequences $x_i^{\text{child}} \in \text{children}(x)$, we compute a selection score vector $\mathbf{U}(x_i^{\text{child}}) = \mathbf{R}(x_i^{\text{child}})/N_{\text{visit}}(x_i^{\text{child}})$ that divides the cumulative reward vector $\mathbf{R}(x_i^{\text{child}})$ of a child node to the number of times it was visited $N_{\text{visit}}(x_i^{\text{child}})$. This encourages exploration of child nodes that have not been visited, in addition to high-reward nodes. Then, we select from the Pareto-optimal set of child nodes based on the normalized selection score vector.

$$\mathcal{P}^*_{\text{select}} = \{x_i^{\text{child}} \mid \nexists x_j^{\text{child}} \in \text{children}(x)$$
$$\text{s.t. } \mathbf{U}(x_j^{\text{child}}) \succ \mathbf{U}(x_i^{\text{child}})\} \tag{6}$$

If we select a non-leaf node, we restart the selection process with the selected node as the new parent until we reach an expandable leaf node or a fully unmasked sequence. In the case of a fully unmasked sequence, we restart the selection from the root node.

**Expansion** From the leaf node $x_t$ at time $t$, we sample $N_{\text{child}}$ distinct sequences by applying Gumbel noise to the denoising distribution $p_{0|t}^\theta(x_0|x_t)$

$$x_0^i \sim \log p_{0|t}^\theta(x_0|x_t) + g_i \tag{7}$$

where $g_i \sim \text{Gumbel}(0, 1)$. Then, for each child node $x_i^{\text{child}}$, we randomly select $k$ positions in $x_t$ to unmask according to $x_1^i$.

During sampling, we employ constraints to ensure that an invalid mRNA sequence is not produced during generation. Specifically, we do not allow codon tokens to be sampled in the position of a UTR token. Additionally, for any masked codon token, we permit only codons encoding for the same amino acid as the corresponding codon in the input template to be sampled. This constraint guarantees that the sampled mRNA sequence does not form an unintended protein product.

**Rollout** For each child node $x_i^{\text{child}}$, we use self-planned ancestral sampling to obtain a fully unmasked sequence $\tilde{x}_i^{\text{child}}$. Then, we feed each of the clean sequences into the vector score function $\mathbf{f}(\tilde{x}_i^{\text{child}})$ to obtain a score vector. We then compute the reward vector $\mathbf{r}(x_i^{\text{child}}) \in \mathbb{R}^K$ of each child $x_i^{\text{child}}$ where each element is the fraction of sequences in the Pareto-optimal set $x^* \in \mathcal{P}^*$ that $x_i^{\text{child}}$ dominates.

$$r_k(x_i^{\text{child}}) = \frac{1}{|\mathcal{P}^*|} \sum_{x^* \in \mathcal{P}^*} \mathbf{1}\left[f_k(x_i^{\text{child}}) \geq f_k(x^*)\right] \tag{8}$$

For all $N_{\text{child}}$ children sequences, we add $x_i^{\text{child}}$ to $\mathcal{P}^*$ if it is *non-dominated* by all sequences $x^* \in \mathcal{P}^*$ and remove all sequences in $\mathcal{P}^*$ dominated by some $x_i^{\text{child}}$.

**Backpropogation** For each child node $x_i^{\text{child}}$, we use its reward vector $\mathbf{r}(x_i^{\text{child}})$ to initialize the cumulative reward $\mathbf{R}(x_i^{\text{child}}) = \mathbf{r}(x_i^{\text{child}})$ and the number of visits $N_{\text{visit}}(x_i^{\text{child}})$ is set to 1.

We then trace backward from $x_i^{\text{child}}$ through its ancestors up to the root node $x_0^{\text{root}}$, updating each node $x$ along the path by accumulating the child rewards and incrementing the visit count:

$$\mathbf{W}(x) \leftarrow \mathbf{W}(x) + \sum_{i=1}^{N_{\text{child}}} \mathbf{r}(x_i^{\text{child}}) \tag{9}$$

$$N_{\text{visit}}(x) \leftarrow N(x) + 1 \tag{10}$$

These cumulative updates influence future selection, prioritizing unmasking paths that lead to higher-reward sequences for further expansion. After $N_{\text{iter}}$ iterations of these four steps, we return the set of Pareto-optimal mRNA sequences optimized across the set of properties $x^* \in \mathcal{P}^*$ and their corresponding score vectors $\mathbf{f}(x^*)$.

---

**Algorithm 2** Monte Carlo Tree Guidance Algorithm (Tang et al., 2025a)

---

1: **Input:** Denoising model $p_\theta(x_1|x_t, t)$, score function $\mathbf{f}(x) : \mathcal{V}^L \to \mathbb{R}^K$, number of iterations $N_{\text{mcts}}$, number of children $N_{\text{child}}$
2: **Output:** Set of Pareto-optimal mRNA sequences $\mathcal{P}^*$
3: **Initialize:** $x_0 \leftarrow [M]^L, \mathcal{P}^* \leftarrow \{\}, t \leftarrow 1$
4: **for** $i = 1$ to $N_{\text{mcts}}$ **do**
5:      **Selection:** Traverse tree to expandable leaf
6:      $x_{\text{leaf}} \leftarrow \text{SELECT}(x_0)$
7:      **Expansion:**
8:      **for** $j = 1$ to $N_{\text{child}}$ **do**
9:          Sample $x_1^{(j)} \sim p_\theta(\cdot|x_{\text{leaf}}, t)$ with Gumbel noise
10:          Create child $x_{\text{child}}^{(j)}$ by unmasking $k$ positions according to $x_1^{(j)}$
11:          Add $x_{\text{child}}^{(j)}$ to children($x_{\text{leaf}}$)
12:      **end for**
13:      **Rollout:**
14:      **for** each $x_{\text{child}}^{(j)} \in$ children($x_{\text{leaf}}$) **do** Complete sequence using self-planned ancestral sampling
15:          $\tilde{x}^{(j)} \leftarrow \text{SELFPLANNEDSAMPLING}(x_{\text{child}}^{(j)})$ Evaluate properties using trained regressors
16:          $\mathbf{s}^{(j)} \leftarrow \mathbf{f}(\tilde{x}^{(j)})$
17:          Update $\mathcal{P}^*$ with $\tilde{x}^{(j)}$ if non-dominated Compute reward based on Pareto dominance (Equation 12)
18:          $\mathbf{r}^{(j)} \leftarrow \text{COMPUTEREWARD}(\mathbf{s}^{(j)}, \mathcal{P}^*)$
19:      **end for**
20:      **Backpropagation:** Update ancestor node statistics with child rewards
21:      $\text{BACKPROPAGATE}(x_{\text{leaf}}, \{\mathbf{r}^{(j)}\})$
22: **end for**
23: **return** $\mathcal{P}^*$

---

Table A2: Hyperparameters for T3PO-mRNA

| Hyperparameter | Value |
|---|---|
| Number of Children | 24 |
| Number of Iterations | 64 |
| Exploration Constant | 0.1 |
| Buffer Size | 64 |
| Number of Diffusion Steps | 64 |

## A.6 HIERARCHICAL CLUSTERING

To evaluate the diversity of generated libraries, we use the SciPy library to perform a hierarchical clustering of RNA secondary structures. To this end, we predict secondary structures for each RNA within the library using mxFold2 (Sato et al., 2021) and subsequently compute a pairwise edit distance matrix for each string in the library via the *linkage* function. Lastly, we employ the *hierarchy* package within SciPy to execute the clustering, with a cluster threshold set to an edit distance of 30.

This was performed for the 5' and 3'UTRs generated by the T3PO-mRNA algorithm, as shown in Figure A2.

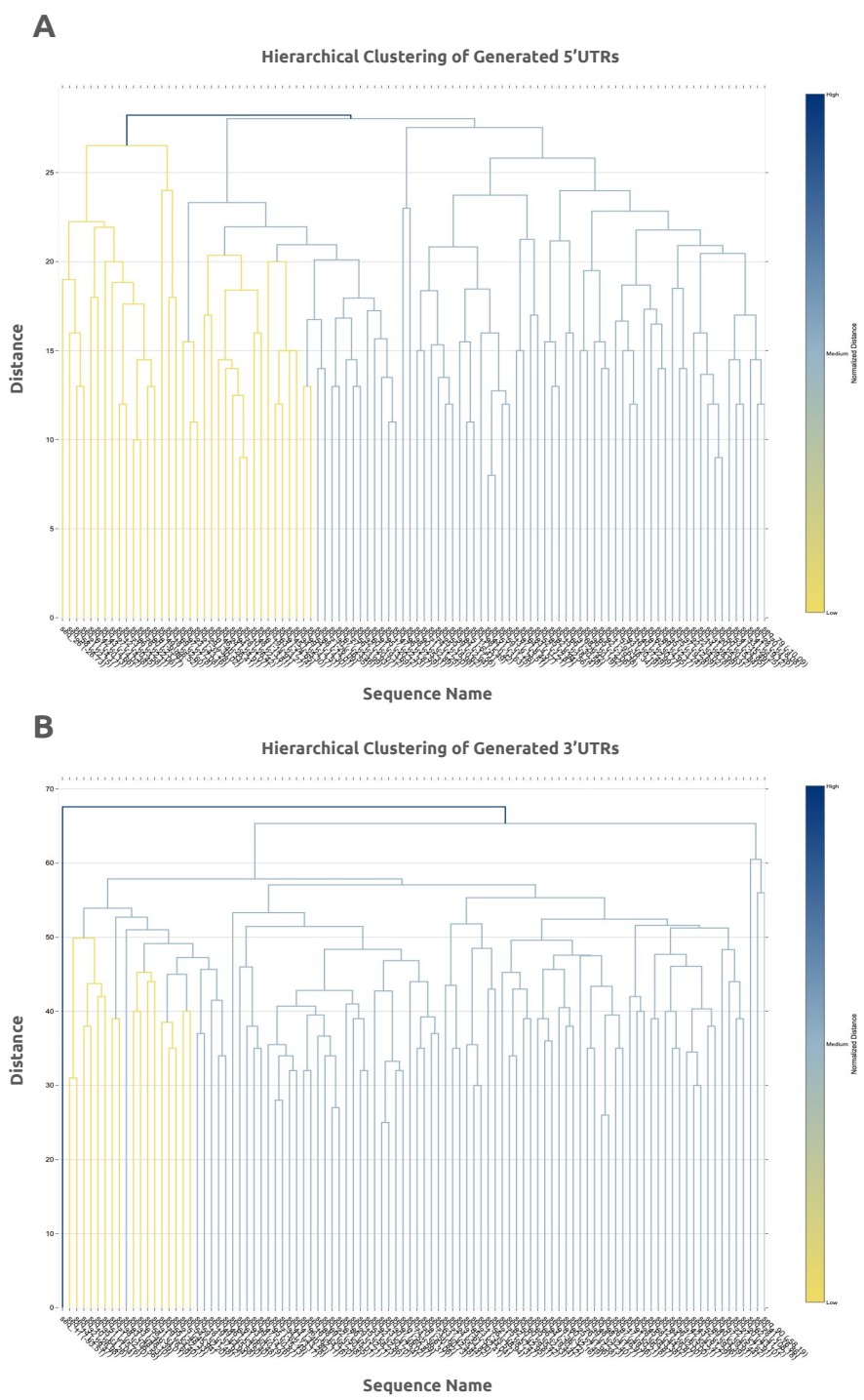

Figure A2: **Hierarchical clustering of generated UTR predicted secondary structures.** UTRs were isolated from generated mRNA sequences according to the prompt specification. Secondary structures were predicted and converted to dot-bracket notation, with pairwise edit distances accordingly computed. Analysis is shown for (**A**) 5' and (**3'**) UTRs.

