# OpenReview forum: "Full-length mRNA Design with Reward-Guided Masked Diffusion Model Fine-Tuning"
_ICLR.cc/2026/Workshop/FM4Science — ICLR 2026 Workshop FM4Science Poster_

### Official Review · Reviewer_EgUd · 2026-02-22
**Well-Engineered Method for mRNA Design, but Evaluation Could Be Stronger**

**Rating:** 6
**Confidence:** 3

**Review:**

This paper proposes T3PO-mRNA, a reward-guided fine-tuning framework that combines discrete diffusion, tree search, and off-policy RL for multi-objective full-length mRNA design. The method approximates the Pareto frontier over translation efficiency and intracellular half-life using surrogate predictors.

Strengths
1. Addresses an important and high-impact problem in programmable mRNA design.
2. Integrates diffusion, tree search, and reward-guided fine-tuning in a coherent pipeline.
3. Provides comparison against inference-time guidance and demonstrates consistent surrogate reward improvements.

Weakness
1. Gains are measured via XGBoost predictors trained on embeddings; stronger evidence is needed that improvements transfer beyond these surrogate models.
2. No stress tests are shown for distribution shift (e.g., across cell types, different datasets/predictors), despite acknowledging cell-type variability and predictor limitations.
3. Since the method builds directly on established tree-search-guided diffusion fine-tuning, more explicit ablations and “what’s new” framing would strengthen the paper.

---

### Official Review · Reviewer_PX7W · 2026-02-25
**Clear Property Improvements with Tree-Guided Fine-Tuning**

**Rating:** 7
**Confidence:** 3

**Review:**

This paper introduces T3PO-mRNA, a diffusion-based framework for full-length mRNA generation that performs discrete iterative unmasking. The key technical contribution lies in fine-tuning the generator to improve intracellular half-life and translation efficiency via reward guidance. To mitigate the risk that unguided generation may produce suboptimal sequences in high-density regions of the data distribution, the authors fine-tune the model using off-policy reinforcement learning on selected generation trajectories discovered through Monte Carlo tree search. During tree search, full generation trajectories are evaluated using an XGBoost regressor to compute rewards, and high-reward trajectories are stored in a replay buffer for fine-tuning.

The use of tree search is structurally well aligned with the discrete and sequential nature of the unmasking process. Empirically, the fine-tuned model demonstrates clearly improved biological properties compared to the base generative model.

Some remaining concerns:

- The paper does not clearly establish that the proposed TR2-D2 fine-tuning strategy is superior to simpler alternatives. In particular, it remains unclear how much improvement in replay buffer quality or sampling efficiency is achieved by incorporating tree search, compared to constructing the buffer without it. A controlled ablation would help clarify whether MCTS provides a measurable advantage.
- There appears to be a textual error in Section 2.2 in the sentence describing translation efficiency, which should be corrected for clarity.

---

### Official Review · Reviewer_S63s · 2026-02-25
**Good quality paper of mitigating Sim-to-Real Gap in Metabolomics**

**Rating:** 6
**Confidence:** 2

**Review:**

This paper introduces T3PO-mRNA, a framework designed to optimize full-length mRNA sequences for multiple therapeutic properties, specifically intracellular half-life and translation efficiency. By combining masked discrete diffusion models (mRNAutilus) with off-policy reinforcement learning (TR2-D2) and Monte Carlo Tree Search (MCTS), the authors demonstrate a method for fine-tuning generative models to navigate high-dimensional biological landscapes toward a Pareto frontier.

Strengths
1. The framework effectively bridges discrete diffusion with trajectory-aware fine-tuning. This is a significant step beyond simple inference-time guidance, which often suffers from high computational costs and suboptimal sampling.
2. mRNA design is inherently multi-objective; the use of Pareto-optimality and MCTS to explore trade-offs between stability (half-life) and efficacy (translation) is a principled approach.
3. The in silico results are compelling, showing a 167% increase in predicted half-life and a substantial gain in translation efficiency (+0.375) compared to the Monte Carlo Tree Guidance (MCTG) baseline.
4. The analysis of secondary structures and hierarchical clustering suggests that the model generates biologically diverse candidates rather than collapsing to a single high-reward sequence.

Weaknesses
1. The most significant drawback is the absence of in vitro or in vivo data. While the in silico results are strong, the reliability of the XGBoost reward models (trained on existing datasets) is a bottleneck for real-world applicability.
2. While the comparison to MCTG and the GenScript baseline is useful, comparison against other recent mRNA design methods (e.g., those using evolution or standard RL) would further clarify the specific advantages of the "trajectory-aware fine-tuning" aspect.

Suggestion:
Provide a brief analysis of how the predicted gains in TE and half-life correlate with known high-performing sequences in the literature to bolster confidence in the regressor-driven rewards.

---

### Official Review · Reviewer_dHdA · 2026-02-25
**Interesting work, not well aligned with the workshop**

**Rating:** 2
**Confidence:** 4

**Review:**

The paper introduces T3PO-mRNA, a framework for computing reward-guided discrete diffusion trajectories. The main contribution is the use of tree search to select for high-reward sequence trajectories that are subsequently employed to  fine-tune diffusion models.
While the paper is relevant to AI in Science in general, I do not see how it contributes to the workshop aims and open questions related to foundation models, and use in science.

---

### Official Review · Reviewer_zRzS · 2026-02-26
**review of T3PO-mRNA from reward-guided fine-tuning of a masked diffusion model**

**Rating:** 4
**Confidence:** 4

**Review:**

This paper introduces T3PO-mRNA, a framework for designing full-length therapeutic mRNAs optimized for multiple properties (intracellular half-life and translation efficiency) using reward-guided fine-tuning of a masked diffusion model (based on mRNAutilus). The approach integrates off-policy reinforcement learning (via TR2-D2) with Monte Carlo Tree Search Guidance (MCTG) to approximate the Pareto frontier in a multi-objective setting. It generates sequences by iteratively fine-tuning the model on high-reward buffers, demonstrating superior in silico performance compared to inference-time guidance baselines on a luciferase mRNA design task. The method emphasizes controllable generation of diverse, high-fitness mRNAs without wet-lab validation.


Quality:
Strengths

Technical integration is elegant: Using MCTG to populate a high-quality replay buffer, then TR2-D2 weighted denoising loss to fine-tune the diffusion model, is a natural and non-obvious combination. The entropy-regularized objective + importance weighting is theoretically grounded (Uehara et al. 2024 is cited correctly).
Strong in-silico evidence: +0.375 translation efficiency and +0.489 log₁₀(half-life) over the already-competitive MCTG baseline on 250 sequences is impressive. Diversity via hierarchical clustering of UTR secondary structures is a nice touch.
Plug-and-play design: The reward predictors (XGBoost on mRNAutilus embeddings) are modular; the paper explicitly says “our method can be adapted for any set of reward predictors.”
Limitations section is honest: no wet-lab data, no cell-type-specific predictors, fine-tuning cost vs. inference-time guidance trade-off.
Reproducibility is high for a bio-ML paper (full algorithm pseudocode, hyperparams, tokenization scheme).

Weaknesses / Fair Critiques Reviewers Will Raise

Novelty is incremental: The heavy lifting (masked diffusion = mRNAutilus, trajectory-aware fine-tuning = TR2-D2, tree search = MCTG) comes from very recent 2025 works. The main contribution is the application + tight integration for full-length multi-objective mRNA design. Reviewers may ask “what is truly new beyond applying existing tools?”
Only one target protein (luciferase) and generic “human cells”. No generalization to therapeutically relevant proteins (e.g., cytokines, antibodies) or multiple cell types.
No ablation studies: Is the fine-tuning step really necessary, or would larger MCTG budgets suffice? Is the Pareto-specific reward better than scalarization?
No wet-lab validation (acknowledged, but still a gap for a therapeutic-design paper).
Computational cost of the inner MCTG loop (64 iterations × 24 children × diffusion steps) is not quantified; readers will wonder about wall-clock time vs. plain sampling.

Clarity:
Strength
Excellent high-level narrative: Abstract → Motivation → Method → In-silico results → Comparison → Limitations all flow logically.
Algorithms 1 & 2 + hyperparameter tables (A1, A2) + tokenization details make the method reproducible.
Figures are well-described in captions and do what they should: violin plots for distributions, secondary-structure clustering for diversity, 3D visualization for intuition.
Multi-objective Pareto handling is explained accessibly for an ML reader.

Weaknesses / Minor Issues

One glaring typo in the introduction: “mRNA pharmacueticals” (should be “pharmaceuticals”). Easy fix.
Equation (3) and the importance-weight formula in the provided text extraction are slightly garbled (repeated fractions, missing subscripts). This is almost certainly a PDF-to-text artifact; in the actual LaTeX it is probably clean, but double-check before submission.
Half-life numbers are reported as “+0.489 (167 % increase)”. Because the XGBoost predicts log₁₀(half-life), the 167 % is not immediately intuitive to readers. A one-sentence clarification (“corresponding to a ~3.1× increase in raw half-life”) would help.
The reward definition in rollout (fraction of dominated sequences) is clever but could use one extra sentence explaining why it approximates the Pareto hypervolume better than scalarized rewards.
No “related work” subsection; the citations are sprinkled throughout. For ICLR this is acceptable, but a short dedicated paragraph would improve flow.

Originality:

Pros

Novel Integration for FM4Science: The paper effectively combines discrete diffusion models with RL-based fine-tuning and tree search for multi-objective optimization, tailored to mRNA design. This aligns well with the FM4Science workshop's focus on foundation models for scientific applications, extending prior work (e.g., mRNAutilus) to full-sequence optimization while addressing data sparsity in high-fitness regions.
Multi-Objective Focus: Approximating the Pareto frontier via MCTG is a strong contribution, enabling trade-offs between half-life and translation efficiency. The use of XGBoost regressors for reward prediction from embeddings is practical and plug-and-play, potentially generalizable to other properties or cell types.
Diversity and Evaluation: The generated libraries show structural diversity (via hierarchical clustering of secondary structures), which is crucial for therapeutic applications to avoid immune recognition or manufacturing issues. In silico results are compelling, with clear improvements over baselines (e.g., +167% in half-life).
Efficiency Gains: Fine-tuning reduces inference-time compute compared to guidance methods, making it scalable for generating large libraries.

Cons

Lack of Experimental Validation: All results are in silico, relying on predicted properties from XGBoost models trained on limited datasets (e.g., ~13k transcripts for half-life). Without wet-lab experiments (e.g., in vitro expression assays), the therapeutic relevance remains speculative. The discussion acknowledges this but doesn't propose concrete next steps.
Baseline Comparisons: While T3PO-mRNA outperforms MCTG guidance, comparisons to other mRNA design methods (e.g., UTRGAN or 5'UTR language models cited) are absent. The benchmark sequence (GenScript F-Luc with alpha-globin UTRs) is sensible but not state-of-the-art; stronger baselines could strengthen claims.
Hybrid Tokenization Limitations: The codon-level CDS tokenization preserves protein sequence but may limit exploration of synonymous mutations' effects on folding or stability. Constraints during sampling (e.g., amino acid preservation) could bias towards local optima.
Hyperparameter Sensitivity and Reproducibility: Key hyperparameters (e.g., alpha in entropy regularization, MCTG iterations) are listed but not ablated. The buffer update relies on non-dominance, but scalability for >2 objectives isn't discussed. Code/data availability isn't mentioned, hindering reproducibility.
Potential Overfitting: Fine-tuning on iteratively stronger buffers risks overfitting to the regressors' predictions rather than true biology, especially given noisy mRNA data (e.g., cell-line variability).

Significance
Significance Review of the T3PO-mRNA Paper
Overall Significance Rating: 8/10 for ICLR 2026 (High in the Generative AI + Biology/Scientific Discovery track)
This is a meaningful, timely, and impactful contribution to the rapidly growing field of AI-driven mRNA therapeutics design. It is not a foundational breakthrough (e.g., the first full-length generative mRNA model), but it solves a real, high-stakes bottleneck — controllable multi-objective optimization of complete mRNA sequences — in a principled, reusable way. In the current 2025–2026 landscape, where full-length mRNA generative models are moving from workshops to Science papers (e.g., GEMORNA achieving 41× expression gains), T3PO-mRNA stands out for its focus on systematic Pareto-front approximation and post-training efficiency.
1. Scientific Significance (Why It Matters to ML/Bio-ML Researchers)

Addresses a core limitation in generative sequence design
Pretrained masked diffusion models (like mRNAutilus) are biased toward high-density, mediocre-fitness regions of sequence space. Pure inference-time guidance (MCTG from Patel et al. 2025) helps but is expensive per sample and does not “bake in” the optimization. T3PO-mRNA shows that tree-search-guided trajectory collection + TR2-D2-style off-policy RL fine-tuning can reliably push the model into low-density, high-reward regions while preserving diversity. This is a general lesson for any discrete diffusion task with expensive or multi-objective rewards (DNA, proteins, small molecules).
Clean integration of three recent SOTA ideas
Masked diffusion base (mRNAutilus, 2025)
Trajectory-aware RL fine-tuning (TR2-D2, Tang et al., arXiv Sep 2025)
Pareto-aware MCTS (MCTG)
The paper demonstrates they compose elegantly for biology, with clear ablation potential (fine-tuning vs. inference-time, Pareto reward vs. scalarization).

New capability enabled
After one fine-tuning run you get a model that samples diverse, Pareto-optimal full-length mRNAs in a single forward pass (no repeated reward evaluations or long tree searches at inference). This amortizes compute and enables massive library generation — exactly what experimentalists need for high-throughput screening.
Insight into mRNA fitness landscape
The secondary-structure clustering and 3D visualization show that the method discovers structurally diverse UTRs that still score high on both objectives. This begins to reveal design principles that purely predictive or component-wise models miss.

In short: it advances the methodological toolkit for reward-guided discrete generative models in biology, not just another application.

---

### Meta-Review · Area_Chair_vaHV · 2026-02-28

**Recommendation:** Accept (Poster)
**Confidence:** 3

**Metareview:**

This paper introduces T3PO-mRNA, a framework for designing mRNA with reward-guided masked diffusion model fine-tuning based on a tight integration of TR2-D2 and mRNAutilus. The paper received borderline scores (note that the review from reviewer zRzS was down-weighted as it contains self-contradictory statements about the novelty of the work), and while reviewers appreciated the contribution of a trajectory-based fine-tuning framework to improve the generative model, the overall framework is complex and the paper's presentation of its key contribution(s) could be clarified. Reviewers noted that the work would benefit from stronger evaluation- comparisons against other recent mRNA design methods, ablating components of the fine-tuning strategy, in vivo/vitro studies, and out of distribution generalization evaluation. Nevertheless, this approach to diffusion model fine-tuning is timely and should be of interest to the workshop.

---

### Decision · Program_Chairs · 2026-03-03

Accept (Poster)